# Development of an instrument to measure mistreatment of women during childbirth through item response theory

**Janini Cristina Paiz**[1]*, **Stela Maris de Jezus Castro**[1,2], **Elsa Regina Justo Giugliani**[3], **Sarah Maria dos Santos Ahne**[4], **Camila Bonalume Dall' Aqua**[1], **Alice Steglich Souto**[1], **Camila Giugliani**[1]

**1** Graduate Program in Epidemiology, Faculty of Medicine, Universidade Federal do Rio Grande do Sul (UFRGS), Porto Alegre, RS, Brazil, **2** Department of Statistics, Universidade Federal do Rio Grande do Sul (UFRGS), Porto Alegre, RS, Brazil, **3** Graduate Program in Child and Adolescent Health, Faculty of Medicine, Universidade Federal do Rio Grande do Sul (UFRGS), Porto Alegre, RS, Brazil, **4** Faculty of Medicine, Universidade Federal do Rio Grande do Sul (UFRGS), Porto Alegre, RS, Brazil

* janinicpaiz@gmail.com

## Abstract

The objective of this study was to structure a proposal for an instrument to measure the mistreatment level of women during childbirth, through item response theory, based on the birth experience of postpartum women. A cross-sectional study was conducted, with the inclusion of 287 women who did not suffer complications during childbirth, randomly selected from two maternity hospitals in the capital of Rio Grande do Sul—Brazil, in 2016. Approximately 30 days after delivery, the women answered questions in a face-to-face interview about their birth experience (practices and interventions applied) and were inquired about their perception of having suffered disrespect, mistreatment or humiliation by health professionals. The set of practices was included in the item response theory model to design the instrument. Of the 36 items included in the model, 21 dealt with practices applied exclusively to women who went into labor, therefore two instruments were developed. The instrument including all women, containing 09 items, identified 23.7% prevalence of mistreatment to women during childbirth, while the instrument for women going into labor included 11 items and identified 22% prevalence. The items with the highest discrimination were: not having had a companion during labor (2.05; and 1.26), not feeling welcome (1.81; and 1.58), and not feeling safe (1.59; and 1.70), for all women and for those who went into labor, respectively. For those who went into labor, the items, did not have a companion during labor (1.22; PE 0.88) and did not feel comfortable asking questions and participating in decisions (1.20; PE 0.43) also showed greater discrimination. In contrast, when directly questioned, only 12.5% of women said they had experienced disrespect or mistreatment, suggesting that harmful practices are often not recognized as violent. Standardizing the measurement of mistreatment of women during childbirth can create more accurate estimates of its prevalence and contribute to the proposal of strategies to eliminate obstetric violence.

**Data Availability Statement:** All relevant data are within the paper and its Supporting Information files.

**Funding:** The source of funding (financial support) for your study was the National Council for Scientific and Technological Development (CNPq). The funders had no role in study design, data collection and analysis, decision to publish, or preparation of the manuscript. No authors received a salary from any of funders.

**Competing interests:** The authors have declared that no competing interests exist.

## Introduction

Childbirth, an essentially physiological event, has over the years become a hospital procedure, with frequent use of routine interventions, often unnecessary and not based on scientific evidence supporting its benefits [1]. Expansion of access to health services, at population level, has provided greater safety to childbirth, which is supervised by qualified health professionals in an environment with technological resources and with the possibility of timely referral, if necessary, to higher complexity services [2, 3].

In 2019, in Brazil, 99.1% of births took place in health facilities and only 43.6% were vaginal [4]. The safety attributed to hospital childbirth care often contrasts with the risks of rigid routines and overburdened professionals in this setting, resulting in increased rates of induced labor, cesarean sections (c-sections), episiotomy, and other interventions that are frequently unnecessary [1, 5]. The combination of these factors impacts the quality of care, making it less compassionate, and hinders the women's experience, often generating the legitimate feeling of mistreatment during childbirth.

A positive birth experience considers and incorporates women's sociocultural beliefs and preferences. It includes the birth of a healthy baby in a safe environment with regard to clinical and psychological issues, with continuity of physical and emotional care, by a qualified technical team [5]. On the other hand, mistreatment refers to the different forms of violence that occur, whether physical, psychological, or verbal, or by unnecessary and harmful procedures, caused to women in prenatal care, childbirth, postpartum, and abortion [1, 6, 7]. The term obstetric violence is more commonly used in Latin America to refer to mistreatment during childbirth, and we agree with its use in the sense it poses a political stance, as it implies the recognition of structural problems that negatively affect women's experiences, jeopardizing the realization of their rights [8]. However, in this paper we opted to use the term mistreatment because of its prevailing use in international scientific literature and also because the actual question made to the women participating in this study (see Materials and methods) contains the term mistreatment, and not the term obstetric violence.

Mistreatment of women during childbirth is considered a global public health problem, and its elimination appears as a priority in women's health policies, drawn up by regulatory bodies, such as the World Health Organization (WHO)—Intrapartum Care for a Positive Childbirth Experience [5]—and the International Federation of Gynecology and Obstetrics (FIGO)—International Childbirth Initiative [9]. In Brazil, over the last 20 years there have been important governmental efforts to improve prenatal, childbirth and postpartum care, such as *Programa de Humanização no Pré-Natal e Nascimento* (Prenatal and Childbirth Humanization Program), instituted in 2000, and *Rede Cegonha* (Stork Network), in 2011. The private sector, with impressively high c-section rates (reaching around 80%) [10], received special attention in 2014, when the *Projeto Parto Adequado* (Appropriate Delivery Project), also called *Nascer Saudável* (Healthy Birth), was launched [11]. These initiatives, led by the Ministry of Health and the National Supplementary Health Agency–ANS, have shown its positive impact on childbirth care in Brazil [6, 11], however they have been weakened in recent years, especially since 2016, due to political changes in the country that have resulted in disinvestment as well as in clear disinterest in the continuity of the mentioned strategies. In this sense, the *Rede Cegonha* has recently lost its name and shape with the official institution of the new *Rede de Atenção Materna e Infantil (RAMI)*–Maternal and Child Care Network [12]. Unlike the first, this new governmental initiative puts marked emphasis on specialized care and high risk situations, while not including traditional birth attendants in any context of childbirth care. In terms of context, it's important to note that the mentioned Childbirth Humanization Program was a governmental response not only to the epidemiological needs linked to

avoidable maternal deaths but also to the demands of social and women's movements claiming for their sexual and reproductive rights. Currently, the new RAMI likely entails risks to the progress achieved in more than 20 years.

A study developed in Brazil, from 2010 to 2013, identified a 25% prevalence in mistreatment of women during childbirth. This prevalence was 50% for women who had abortions [13]. While the research *Nascer no Brasil* (Birth in Brazil), conducted in the same period (2011–2013) with national representation, involving more than 16,000 postpartum women, identified a lower prevalence (5.1%) of mistreatment, abuse or violence by health professionals, as perceived by the women [14].

More recent surveys, such as one conducted in 2019 involving postpartum women from the Brazilian public and private networks, showed reduction in the rates of several interventions considered harmful when performed without precise indication: elective c-sections, episiotomy, application of manual pressure on the uterine fundus, and lithotomy position in labor. At the same time, it highlighted the increase in the prevalence of good practices, such as having a companion chosen by the woman, receiving food during labor, and the use of non-pharmacological methods for pain relief [11].

Even with current awareness of the impacts of mistreatment on quality and satisfaction with care, damaging the health of women and children [15–19], there are gaps in the understanding of this problem. These gaps are due, in part, to the absence of an indicator (or instrument) that measures mistreatment of women during childbirth, built from the perception of those who experience it, as well as the variability in how it is measured, which interferes with the estimation of its prevalence and comparability between studies [14, 18, 20]. Therefore, the aim of this study was to structure a proposal for an instrument to measure the Mistreatment Level of Women during Childbirth (MLWC), through the Item Response Theory (IRT), based on the childbirth experience reported from a sample of postpartum women. The secondary objective was to compare the prevalence of mistreatment achieved through the proposed instrument with the one measured through a specific and direct question applied to this sample of women.

## Materials and methods

### Study design and population

We conducted a cross-sectional study, including postpartum women who gave birth in two large maternity hospitals of Porto Alegre, Rio Grande do Sul (RS). One is a general university hospital, predominantly public, and the other is a private general hospital. Both are centers of excellence and referral institutions for both low and high-risk pregnancies, having performed, in the year 2016, 3725 and 4182 deliveries (26% of the city's total), respectively. Reflecting the national trend, the private hospital stands for a much higher c-section rate (81%, compared with 33% in the public hospital) [4, 21]. In both facilities, deliveries are mainly assisted by obstetricians; the inclusion of midwives in childbirth care is recent and still incipient. Only the public facility, as a teaching hospital, includes medical and nursing students and residents in maternity care.

All women living in Porto Alegre who gave birth to full-term newborns in the two participating maternity hospitals without unfavorable outcomes at delivery (death or admission to intensive care) and with no formal contraindication for breastfeeding were considered eligible. These eligibility criteria were defined in order to avoid biases in the measure of the main outcomes of interest (satisfaction with childbirth and breastfeeding prevalence) in the research that originated this study [18, 22]. Women living in areas considered dangerous for home visits had to be excluded to preserve the safety of the research team.

## Sample and data collection

The data used in this study derives from former research designed to identify the factors associated with women's satisfaction with childbirth. For this objective, the sample size calculation reached 276 women. More details on the methods and results of the former research have been published elsewhere [18].

Data collection occurred between January and August 2016. Every day, all eligible women who had given birth in the previous 24 hours received a number that was used for a draw. Each day, two women from the public maternity hospital and one from the private hospital were randomly selected and included in the study until the intended sample was reached. This proportion aimed to ensure a reasonable representation in relation to the use of public and private services, described in the literature as being around 70% and 30%, respectively, at national level [23, 24].

In the period from 31 to 37 days after delivery, an interview was conducted at the home or, rarely, in another place at the woman's preference, to apply a structured questionnaire, which was specifically designed for this study, based on the previous experience of the researchers and the guiding documents of childbirth care in Brazil [6, 25]. The questionnaire had an average application time of 55 minutes and included items related to sociodemographic characteristics, women's health and obstetric history, prenatal and childbirth care received. The socioeconomic level was assessed according to the Brazilian Research Enterprises Association, based on the possession of a series of domestic items and on the householder's education level [26]. The grouping of categories from A to E corresponds to a range from better off (A) to worse off (E). Name, age, education and skin color were drawn from hospital records, while all other information considered in this study was collected during face-to-face interviews and the responses were referred by the participants. Women who were not found for the interview, after at least three attempts of contact by telephone and one in person, were considered a loss.

The interviews were carried out after a pilot project had been conducted and adjustments to the questionnaire had been made. The field team was composed of 12 interviewers trained for the job. Weekly meetings were held with the field team, seeking greater uniformity in data collection.

Some potential sources of bias were minimized with randomization (selection bias), face-to-face interviews (data quality), team training for data collection and weekly monitoring (measurement bias), and timing of interviews (1 month after delivery, not too short nor too long, avoiding at the same time gratitude bias and memory bias, respectively). Data was collected manually using paper copies of the questionnaires and then entered by two independent researchers, enabling reliable verification of the database. Moreover, responses to some key questions of the questionnaire were double-checked by telephone with 5% of the participants, selected by lot.

## Statistical aspects

In this study, two different ways were used to measure mistreatment. First, through a direct question, to assess the woman's perception of having suffered disrespect, mistreatment or humiliation, using the following question: *Have you ever (during labor and childbirth care) felt disrespected, humiliated or mistreated by health professionals*? having as response alternatives: yes, no and don't know/do not remember. The second consisted in creating a measurement instrument for the latent trait mistreatment level of women during childbirth (MLWC) from a survey on childbirth experience, practices and interventions applied to the sample of women included in the study.

To create the measurement instrument, the IRT model known as the Two-Parameter Logistic Model was adapted [27, 28]. In the context of this study, the model predicts probability that a woman $j$, with MLWC $\theta j$, will respond to category 1 of item $i$ (items presented in Table 1), i.e., $P(Xij = 1 \mid \theta j)$, as follows:

$$P\left(X_{ij} = 1|\theta_j\right) = \frac{1}{1 + e^{-Da_i(\theta_j - b_i)}}$$

being $i = 1,2,\cdots,11$ and $j = 1,2,\cdots,n$,

where $X_{ij}$ is the dichotomous item that takes the values 1 or 0;

$\theta j$ represents the MLWC of the $j$th woman, estimated on the scale with mean zero and standard deviation 1;

$b_i$ is the position parameter of item $i$, estimated on the same scale as $\theta$, which represents the severity of the content measured in item $i$, that is, the level of mistreatment necessary for the woman to answer category 1 of item $i$ with probability equal to 0.5;

$a_i$ is the discrimination (or slope) parameter of item $i$—low values of this parameter indicate that women with different levels of mistreatment are about equally likely to answer category 1 of item $i$, and very high values of this parameter discriminate women basically into two groups: the group that has a level of mistreatment below the value of the parameter $bi$ and the group that has a level of mistreatment above the value of the parameter $bi$;

$D$ scale factor (constant, equal to 1 or 1.7).

One of the products of the IRT models are the so-called item characteristic curves (ICC). The ICCs of the two-parameter logistic model used in this study describe how changes in MLWC are related to changes in the probability of response in category "1" of each item of the measurement instrument. Items with high discrimination values (it is desirable for items to have estimates for the parameter $a_i$ greater than or equal to 1) have the steepest ICCs [27].

Another product of the IRT models are the item information curves (IIC) and test information curves (TIC). The IICs describe the amount of psychometric information that each item adds to the estimate of MLWC for different levels of that latent trait. The TIC is a sum of the IICs and shows for which MLWC the created measurement instrument estimates the most accurate latent trait.

The two-parameter logistic model must satisfy two assumptions: local independence and one-dimensionality (a single latent trait is determining the responses to the items). These assumptions are related, so that if the measurement instrument is unidimensional, local independence is met [27, 29]. The assumption of one-dimensionality can be made flexible, which is known as sufficient one-dimensionality, that is, it is enough that there is a preponderant dimension (explanation proportion of the first dimension at least equal to 20%) so that the model can be used [30–32]. Verification of sufficient one-dimensionality of the proposed measurement instrument was performed by means of exploratory factor analysis, using the tetrachoric correlation matrix, since the items are dichotomous responses.

The cut-off point of the IRT score for defining mistreatment was 0.5 standard deviation above the mean. The binomial test was used to compare the prevalence of MLWC with the prevalence obtained by the perception of violence, measured through a direct question.

To create the MLWC measure, the *ltm*, version 1.1–1, and *psych*, version 2.1.6 packages were used. The remaining analyses were performed using the SPSS software, version 18.

## Ethical aspects

This study complies with the standards governing research with human subjects [33] and was approved by the research ethics committees of the institutions involved (CAAE

**Table 1. Characteristics of the sample of postpartum women regarding sociodemographic aspects, lifestyle, reproductive history, prenatal and childbirth care, according to the perception of mistreatment through the question:** *Have you ever felt disrespected, humiliated or mistreated by health professionals?* **Porto Alegre, 2016.**

| | Sample n (%) | Felt disrespected, mistreated, or humiliated–n (%) | |
|---|---|---|---|
| **Predictor variables** | **n = 287** | **Yes n = 36–12.5%** | **No n = 251–87.5%** |
| **Sociodemographic** | | | |
| **Age (year)** | | | |
| ≤ 19 years | 23 (8.0) | 2 (8.7) | 21 (91.3) |
| 20–34 years | 199 (69.3) | 32 (16.1) | 167 (83.9) |
| ≥35 years | 65 (22.6) | 2 (3.1) | 63 (96.9) |
| **Color of skin** | | | |
| White | 216 (75.3) | 28 (13.0) | 188 (87.0) |
| Black or brown | 71 (24.7) | 8 (11.3) | 63 (88.7) |
| **Socioeconomic level (n = 285)**\* | | | |
| A–B | 163 (57.2) | 18 (11.0) | 145 (89.0) |
| C–D–E | 122 (42.8) | 18 (14.8) | 104 (85.2) |
| **Education** | | | |
| College | 124 (43.2) | 15 (12,1) | 109 (87.9) |
| Elementary and high school | 163 (56.8) | 21 (12.9) | 142 (87.1) |
| **Lives with a partner** | | | |
| Yes | 248 (86.4) | 33 (13.3) | 215 (86.7) |
| **Lifestyle and Health Status** | | | |
| **Smoker** | | | |
| Current or past | 92 (32.1) | 11 (12.0) | 81 (88.0) |
| **Mental health condition** | | | |
| Current or past | 38 (13.2) | 6 (15.8) | 32 (84.2) |
| **Use of psychoactive medication** | | | |
| Current or past | 55 (19.2) | 9 (16.4) | 46 (83.6) |
| **Reproductive history** | | | |
| **Previous births** | | | |
| One | 142 (49.5) | 21 (14.8) | 121 (85.2) |
| Two | 98 (34.1) | 7 (7.1) | 91 (92.9) |
| Three or more | 47 (16.4) | 8 (17.0) | 39 (83.0) |
| **Miscarriage history (n = 163)**\*\* | | | |
| Yes | 47 (28.8) | 5 (10.6) | 42 (89.4) |
| **Last planned pregnancy** | | | |
| Yes | 154 (53.7) | 19 (12.3) | 135 (87.7) |
| **Prenatal Care** | | | |
| **Number of visits (n = 282)**\* | | | |
| ≤ 7 appointments | 46 (16.3) | 4 (8.7) | 42 (91.3) |
| 8 or more | 236 (83.7) | 31 (13.1) | 205 (86.9) |
| **Companion of her choice** | | | |
| Yes, at least one appointment | 218 (76.0) | 31 (14.2) | 187 (85.8) |
| **Informed about her rights (n = 282)**\* | | | |
| Yes, totally | 152 (53.9) | 15 (9.9) | 137 (90.1) |
| **Had a birth plan** | | | |
| Yes | 15 (5.2) | 1 (6.7) | 14 (93.3) |
| **Felt free to ask questions** | | | |
| Yes, totally | 241 (84.0) | 26 (10.8) | 215 (89.2) |
| **Childbirth care** | | | |

*(Continued)*

**Table 1.** (Continued)

| | Sample n (%) | Felt disrespected, mistreated, or humiliated–n (%) | |
|---|---|---|---|
| **Predictor variables** | **n = 287** | **Yes n = 36–12.5%** | **No n = 251–87.5%** |
| **Hospital status** | | | |
| Public | 188 (65.5) | 26 (13.8) | 162 (86.2) |
| Private | 99 (34.5) | 10 (10.1) | 89 (89.9) |
| **Had to go to more than one maternity hospital** | | | |
| Yes | 29 (10.1) | 4 (13.8) | 25 (86.2) |
| **Had a companion** | | | |
| During labor | 275 (95.8) | 34 (12.4) | 241 (87.6) |
| Delivery | 283 (98.6) | 36 (12.7) | 247 (87.3) |
| Postpartum | 275 (95.8) | 34 (12.4) | 241 (87.6) |
| **Felt comfortable asking questions (n = 284)*** | | | |
| Yes | 241 (84.9) | 24 (10.0) | 217 (90.0) |
| **Understood information received** | | | |
| Yes | 251 (87.5) | 28 (11.2) | 223 (88.8) |
| **Went into labor** | | | |
| Yes | 205 (71.4) | 30 (14.6) | 175 (85.4) |
| **Had skin-to-skin contact with the baby (n = 281)*** | | | |
| Yes, immediately after delivery | 167 (59.4) | 23 (13.8) | 144 (86.2) |
| Yes, after performance of procedures on baby | 24 (8.5) | 3 (12.5) | 21 (87.5) |
| **Felt welcomed in the environment (n = 281)*** | | | |
| Yes | 220 (78.3) | 23 (10.5) | 197 (89.5) |
| **Felt safe in the environment (n = 282)*** | | | |
| Yes | 209 (74.1) | 24 (11.5) | 185 (88.5) |
| **Had privacy (n = 280)*** | | | |
| Yes | 235 (83.9) | 22 (9.4) | 213 (90.6) |
| **Care provided to women who went into labor** | **n (%)** | **Felt disrespected, mistreated, or humiliated–n (%)** | |
| | **205 (71.4)** | **Yes 30 (14.6)** | **No 175 (85.4)** |
| **Used pain relief methods** | | | |
| Yes | 168 (82.0) | 28 (16.7) | 140 (83.3) |
| No | 37 (18.0) | 2 (5.4) | 35 (95.6) |
| **Analgesia requested but not received (n = 202)*** | | | |
| Yes | 31 (15.3) | 9 (29.0) | 22 (71.0) |
| No | 171 (84.7) | 20 (11.7) | 151 (88.3) |
| **Offered liquids and light foods (n = 204)*** | | | |
| Yes | 113 (55.4) | 19 (16.8) | 94 (83.2) |
| No | 91 (44.6) | 10 (11.0) | 81 (89.0) |
| **Encouraged to walk (n = 204)*** | | | |
| Yes | 88 (43.1) | 14 (15.9) | 74 (84.1) |
| No | 116 (56.9) | 16 (13.8) | 100 (86.2) |
| **Trichotomy (n = 203)*** | | | |
| Yes | 12 (5.9) | 0 (0.0) | 12 (100) |
| No | 191 (94.1) | 29 (15.2) | 162 (84.8) |
| **Enema (n = 203)*** | | | |
| Yes | 7 (3.4) | 0 (0.0) | 7 (100) |
| No | 196 (96.6) | 29 (14.8) | 167 (85.2) |
| **Induction with oxytocin (n = 193)*** | | | |
| Yes, with or without consent | 108 (56.0) | 21 (19.4) | 87 (80.6) |

*(Continued)*

**Table 1.** (Continued)

| | Sample n (%) | Felt disrespected, mistreated, or humiliated–n (%) | |
|---|---|---|---|
| **Predictor variables** | **n = 287** | **Yes n = 36–12.5%** | **No n = 251–87.5%** |
| Yes, without consent | 16 (8.3) | 12 (75.0) | 4 (25.0) |
| **Amniotomy (n = 201)\*** | | | |
| Yes, with or without consent | 98 (48.8) | 14 (14.3) | 84 (85.7) |
| Yes, without consent | 19 (9.5) | 7 (36.8) | 12 (63.2) |
| **Fundal pressure maneuver (n = 200)\*** | | | |
| Yes | 22 (11.0) | 4 (18.2) | 18 (81.8) |
| No | 178 (89.0) | 24 (13.5) | 154 (86.5) |
| **Episiotomy (n = 204)\*** | | | |
| Yes, with or without consent | 72 (35.3) | 11 (15.3) | 61 (84.7) |
| Yes, without consent | 21 (10.3) | 3 (10.3) | 18 (85.7) |
| **Forceps (n = 200)\*** | | | |
| Yes, with or without consent | 11 (5.5) | 1 (9.1) | 10 (90.9) |
| Yes, without consent | 5 (2.5) | 1 (20.0) | 4 (80.0) |
| **Would like another position in labor (n = 151)\*** | | | |
| Yes | 11 (7.3) | 5 (45.5) | 6 (54.5) |
| No | 140 (92.7) | 20 (14.3) | 120 (85.7) |

\*Missing data correspond to responses "I don't know" or "I don't remember".

\*\*Missing data corresponds to women that were not applicable for this response.

49938015.3.0000.5327 and 46775115.0.3002.5330). All women who agreed to participate in the study signed an informed consent form.

## Results

Among the postpartum women selected for this study, 379 were eligible. Of these, 287 were effectively interviewed. There were 25 (6.6%) refusals, and 67 women (17.7%) were lost due to contact failure to schedule the interviews (data in S1 Fig). The women not interviewed due to losses and refusals differed in terms of education and skin color, showing less education (p<0.01) and a higher prevalence of white skin color compared to those interviewed (p = 0.032).

To develop an instrument capable of measuring the MLWC, 36 items (variables) were initially included, considered as observable facets of the latent trait. As 21 items referred only to women who had gone into labor, including those who ended up having a c-section, we opted for the development of two measures, with the purpose of exploring the particularities related to mistreatment in each of the groups: for all women and for women who went into labor.

Due to the occurrence of missing data, lack of variability in responses and internal consistency measures (Crombach's Alpha), several items were eliminated in the exploratory phase for the development of the instruments for measuring the two MLWC (data in S1 Table). For instance, episiotomy and amniotomy, which were highly prevalent in our sample (35.3% and 48.8% amongst the 205 women who went into labor, respectively–Table 1), were eliminated because they did not contribute psychometric information in the estimation of the latent trait.

Due to the sample size and the low frequency of response in specific categories, the items (variables) were dichotomized (data in S2 Table). In the final model, the measurement instrument covering all postpartum women consisted of nine items (MLWC1), while the measurement instrument covering only women who went into labor consisted of 11 items (MLWC2)

—data in S2 Table. The assumption of sufficient one-dimensionality was met in both measures. For MLWC1, the first factor explained 32% of the total variance among the nine items of the measure, and for MWC2, the first factor explained 31% of the variance among the 11 items of the measure.

The prevalence of mistreatment was 12.5% and 23.7% (p<0.001), for perceived disrespect, mistreatment or humiliation, measured through the direct question; and for the MLWC, constituted through a set of variables, respectively. When only the women who went into labor were considered, these proportions were 14.6% and 22.0% (p = 0.002), respectively. Table 1 shows the characteristics of the sample, which was composed mostly of women aged 20–34 years, white, with a partner and with frequent prenatal care (more than eight visits). Regarding the place of delivery, 21.7% of women did not feel welcome in this environment, 25.9% did not feel safe, and 16.1% reported having no privacy.

Of the 287 women studied, 71.4% went into labor. The induction or acceleration of labor with oxytocin occurred in more than half of the women, similar prevalence to the occurrence of amniotomy, and about 10% of these women did not consent to these procedures. As for the delivery position, 99.3% gave birth in the lithotomy position, and only 7.3% of the women interviewed said they would like other positions. Exposure of women to fundal pressure maneuvers occurred in 11.0% of deliveries (Table 1).

Forty-eight percent of the women in our sample went through a c-section and almost 29% did not initiate labor (were submitted to an elective c-section). Table 2 presents c-section rates according to key characteristics: hospital status (public x private), women's age, skin color, education level, socioeconomic level, parity, and feeling mistreated, disrespected or humiliated. Higher c-section rates were associated with private hospital, as well as with older, white and highly educated women, with a less vulnerable socioeconomic level. The opposite associations were found for women who went into labor. C-section rates were lower amongst women who did not feel mistreated.

Table 3 describes the severity and discrimination measures for each item analyzed by IRT. For MLWC1, the items with the highest severity were the following: not having had a companion during labor (5.18; standard error—SE 2.95), delivery (3.01; SE 0.79), postpartum (4.47; SE 2.13), and not having understood information given by health professionals (3.11; SE 1.15).

For the MLWC1, the characteristic curves (ICC) and the information curves (IIC) of the items 'No companion at delivery' and 'No privacy at delivery' show that the higher the MLCW, the greater the likelihood of the woman answering that she had no companion at delivery and no privacy at delivery (Fig 1). The difference is in the slope of the characteristic curves of the two items, with the item 'No privacy at delivery' having the less steep curve, because this item has a much lower discrimination ability (0.73—Table 3) than the other item (2.05—Table 2). The two IICs reflect this difference. They present the amount of information that each item provides to the estimation of the latent trait MLWC1. The item with the higher discrimination ability has its peak at a higher amount of information (approximately at 0.9) than the other item, which has its peak at a lower amount of information (approximately 0.13).

Fig 2 presents the standard error of the MLWC1 measure, i.e., the precision of the estimate of the latent trait MLWC1 for any level of this trait. It can be seen that postpartum women whose MLWC1 value is in the range of approximately 0.5 standard deviation above the mean will have more accurate estimates of this trait than those whose MLWC1 is below this value. This means that the proposed measurement instrument performs better in postpartum women with higher mistreatment levels.

For MLWC2 (women who went into labor, Table 4), the items with the highest severity were as follows: not having had a companion in the postpartum period (4.98; PE 3.18), having

**Table 2. C-section rates and proportion of women who went into labor according to key characteristics (n = 287).** Porto Alegre, 2016.

| | C-section | p-value* | Went into labor | p-value* |
|---|---|---|---|---|
| | n = 138 (48.1%) | | n = 205 (71.4%) | |
| | n (%) | | n (%) | |
| **Hospital status** | | <0.001 | | <0.001 |
| Public | 60 (31.9) | | 164 (87.2) | |
| Private | 78 (78.8) | | 41 (41.4) | |
| **Age (year)** | | <0.001 | | <0.001 |
| < 35 years | 89 (40.1) | | 172 (77.5) | |
| ≥35 years | 49 (75.4) | | 33 (50.8) | |
| **Color of skin** | | 0.056 | | 0.015 |
| White | 111 (51.4) | | 146 (67.6) | |
| Black or brown | 27 (38.0) | | 59 (83.1) | |
| **Education level** | | <0.001 | | <0.001 |
| College | 83 (66.9) | | 66 (53.2) | |
| Elementary and high school | 55 (33.7) | | 139 (85.3) | |
| **Socioeconomic level (n:285)**** | | <0.001 | | <0.001 |
| A–B | 97 (59.5) | | 96 (58.9) | |
| C–D–E | 41 (33.6) | | 107 (87.7) | |
| **Parity** | | 0.409 | | 0.191 |
| Primiparous | 72 (50.7) | | 96 (67.6) | |
| Multiparous | 66 (45.5) | | 109 (75.2) | |
| **Feeling of mistreatment** | | 0.032 | | 0.114 |
| Yes | 11 (30.6) | | 30 (83.3) | |
| No | 127 (50.6) | | 175 (69.7) | |

*Fischer's Exact test.

**Missing data correspond to responses "I don't know" or "I don't remember".

undergone the fundal pressure maneuver (4.84; PE 3.19), and not having had a companion during labor (4.40; PE 2.37) (Table 4).

With regard to discrimination, both in the general sample of women and for the group of those who went into labor, it was higher for the following items: not having had a companion during labor (2.05 general sample; 1.26 went into labor), not feeling welcomed (1.81 general sample; 1.58 went into labor), and not feeling safe in the maternity environment (1.59 general sample; 1.70 went into labor). In the group that went into labor, the items, did not have a companion during labor (1.22; SI 0.88) and did not feel comfortable to ask questions and participate in decisions (1.20; SI 0.43) also showed greater discrimination.

## Discussion

The prevalence of mistreatment of women during childbirth found in this study, measured in two different ways, were 12.5% using a direct question and 23.7% through an instrument containing a set of items (factors). This variability is due to the use of different ways to measure the same phenomenon and reinforces the differences found in the literature regarding the prevalence of mistreatment [13, 14, 20]. In the international scenario, a systematic review with Latin American and Caribbean studies published between 1990 and 2017, identified a prevalence of 43% of disrespect and mistreatment in childbirth care [34] while another, without geographic restriction, found proportions that varied from 15% to 98% [35].

**Table 3. Parameter estimates of items of the measuring instrument for Mistreatment Level of Women during Childbirth (MLWC1) in postpartum women in public and private hospitals (n = 287).**

| Items | Items (variables) | Severity* | Discrimination* |
|---|---|---|---|
| | **Did not have a companion** | | |
| 1 | During labor | 5.18 (2.95) | 0.64 (0.41) |
| 2 | In postpartum | 4.47 (2.13) | 0.76 (0.42) |
| | **Understood information given** | | |
| 3 | No or not all | 3.11 (1.15) | 0.68 (0.28) |
| | **Did not have a companion** | | |
| 4 | At delivery | 3.01 (0.79) | 2.05 (1.08) |
| | **Had privacy during birth** | | |
| 5 | Little or not at all | 2.48 (0.81) | 0.73 (0.27) |
| | **Felt comfortable to ask questions and participate in decisions** | | |
| 6 | No or sort of | 2.06 (0.52) | 0.99 (0.32) |
| | **Felt welcomed in the environment** | | |
| 7 | More or less, little or not at all | 1.07 (0.20) | 1.81 (0.56) |
| | **Felt safe in the environment** | | |
| 8 | More or less or not at all | 0.94 (0.19) | 1.59 (0.49) |
| | **Had skin-to-skin contact in the delivery room** | | |
| 9 | No or only after procedures | 0.93 (0.19) | 0.43 (0.19) |

* Point Estimation (Standard Error)

The lack of standardization in the definition of mistreatment of women during childbirth, sometimes also referred as obstetric violence, the different instruments used for its measurement and the methodological weaknesses of the studies introduce potential systematic errors for this measure, aspects that affect the generalization and comparability of estimates [35]. Some authors also suggest that this disparity also occurs due to the non-recognition, by women, of situations considered violent according to the definitions supported by official bodies [19], such as the WHO [5] and the Brazilian Ministry of Health [6], as well as the scientific community [1, 36].

A Brazilian study [19] with over 550 postpartum women showed that 51.7% of them categorized their knowledge about obstetric violence as none, little or reasonable. In this study, 12.6% of women reported having experienced this kind of violence, and this situation was associated with marital status (single/separated women), lower income, delivery in lithotomy position, performance of the pressure maneuver on the uterine fundus, and separation from the newborn soon after birth [19].

The prevalence of non-recommended practices found in the present study is similar to those of another investigation, which identified that lithotomy position was present in 91.7% of deliveries, that oxytocin infusion and amniotomy were used in 40% of women [19]. Despite evidence showing that upright positions are associated with shorter labor time, less intense pain sensation, less use of interventions and greater satisfaction with the birth experience [37], lithotomy position is still almost universal in hospital childbirth in Brazil. Moreover, the fundal pressure maneuver, a painful procedure that has been shown to be associated with increased risk of uterine rupture, perineal injuries, neonatal fractures and brain injuries [16, 19], had a prevalence of 37% in another study [19], higher than that found in the present research (11.0%).

Our study identified that more than one third of women who went into labor were submitted to an episiotomy. The high prevalence of this procedure was also observed in the survey

### CCI-No companion at delivery

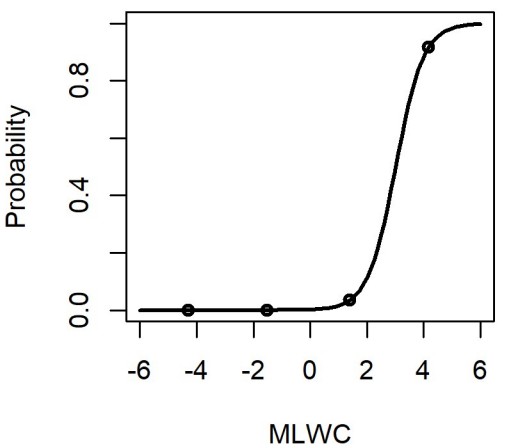

### CII-No companion at delivery

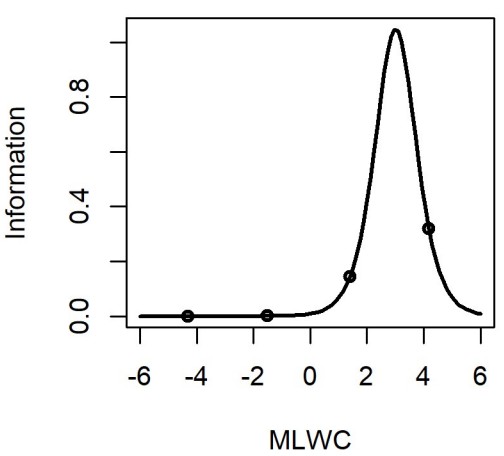

### CCI-No privacy at delivery

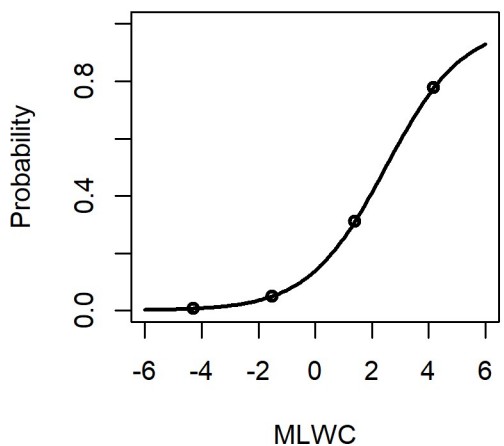

### CII-No privacy at delivery

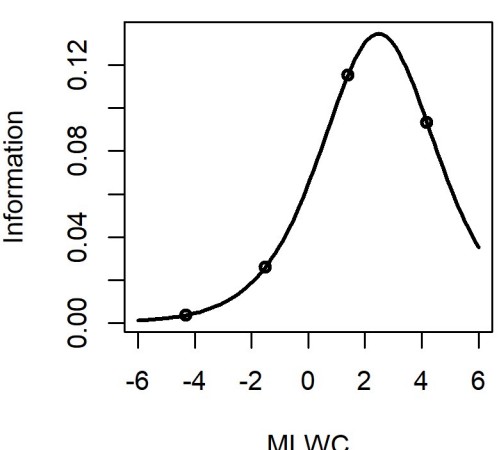

**Fig 1. Characteristic and information curves of the items "no companion" and "no privacy" at delivery for the MLWC1 model (n: 287).**

Birth in Brazil (50%), with a higher occurrence in primiparous women (75%) [36]. Routine episiotomy increases the risk of third- and fourth-degree perineal laceration, infection and bleeding, without reducing the complications of pain and urinary and fecal incontinence in the long term [17]. Unawareness of the risks associated with this procedure and contradictions in its indication cause many women to perceive it as a routine practice. This was seen in the application of the IRT model, in which the inclusion of the variable did not contribute psychometric information, with the caveat that the small sample size may have been responsible for the non-differentiation of this variable in the model. The same might have occurred for other frequent interventions, such as amniotomy, that were eliminated in the exploratory phase (S1 Table).

An interesting finding of the present study is the contrasting rates of c-sections and proportions of women who went into labor according to some key characteristics. Going into labor

# Standard error of measurement

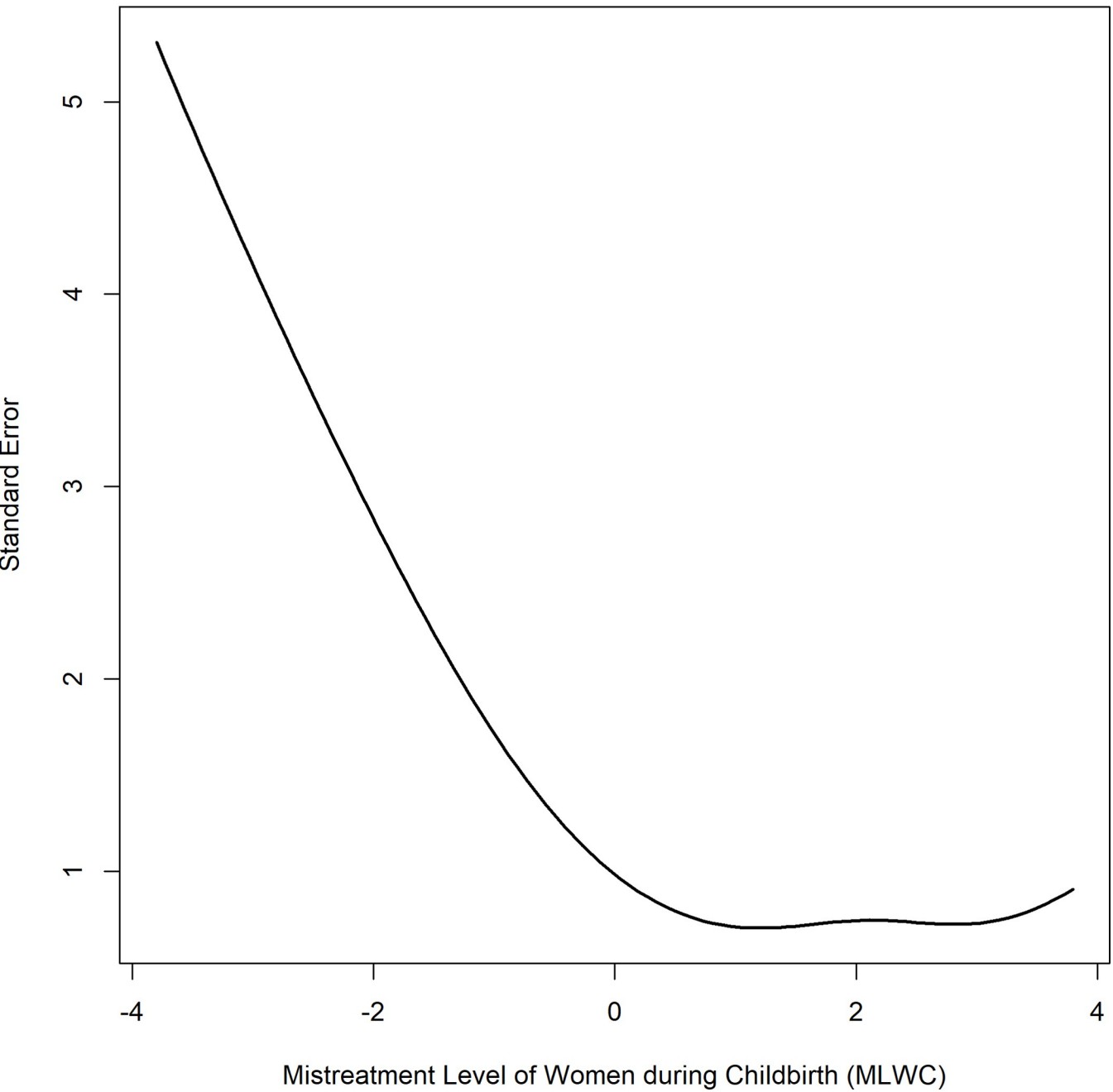

**Fig 2. Standard error for the MLWC1 model (n: 287).**

was more frequent in the public hospital, as well as among younger black or brown women, with lower socioeconomic and education levels. Previous studies have shown that going into labor is a risk factor for mistreatment, as has been evidenced in the research Birth in Brazil: women who went into labor had a 79% higher chance of reporting verbal, psychological or physical violence [15]. It is also relevant to note that c-section rates in our study were higher

**Table 4. Parameter estimates of the items of the measuring instrument for Mistreatment Level of Women during Childbirth (MLWC2) in postpartum women who went into labor (n: 205).**

| Items | Items (variables) | Severity* | Discrimination* |
|---|---|---|---|
| | **Had no companion** | | |
| 1 | In postpartum | 4.98 (3.18) | 0.72 (0.52) |
| | **Underwent fundal pressure maneuver** | | |
| 2 | Yes | 4.84 (3.19) | 0.45 (0.31) |
| | **Did not have companion** | | |
| 3 | At delivery | 4.40 (2.37) | 1.22 (0.88) |
| | **Did not have companion** | | |
| 4 | During labor | 3.36 (1.10) | 1.26 (0.58) |
| | **Understood information given** | | |
| 5 | No or not all | 3.12 (1.37) | 0.64 (0.31) |
| | **Asked for analgesia and was not assisted** | | |
| 6 | Yes | 2.55 (0.93) | 0.74 (0.31) |
| | **Had privacy during delivery** | | |
| 7 | Little or none | 2.53 (1.01) | 0.65 (0.29) |
| | **Had skin-to-skin contact with newborn** | | |
| 8 | No or only after procedures with newborn | 1.57 (0.66) | 0.57 (0.25) |
| | **Felt comfortable to ask questions and participate in decisions** | | |
| 9 | No or more or less | 1.50 (0.40) | 1.20 (0.43) |
| | **Felt welcomed in the environment** | | |
| 10 | More or less, little or nothing | 1.10 (0.24) | 1.58 (0.51) |
| | **Felt safe in the environment** | | |
| 11 | No or more or less | 0.95 (0.22) | 1.70 (0.62) |

* Point Estimation (Standard Error)

among women who did not feel mistreated, which points to the apparent convenience of elective procedures, where women do not initiate labor and the process of care usually flows under the control of both patients and assistant professionals. However, this apparent convenience is highly questioned, once the performance of the surgical procedure without or with incomplete clarification of risks and benefits or even without the woman's deliberate consent is a situation of veiled mistreatment that is usually not recognized as such [38, 39]. These women might feel they were not listened to regarding their preferences, and they commonly feel lonely and vulnerable after the procedure in face of the pain and the limitations to take care of themselves and of the newborn [39]. Women who go through a c-section after previous labor report feeling disrespected more frequently [40], especially due to poor communication with the health team [39].

The items related to the absence of a companion during labor, delivery and postpartum periods were the most important for the MLWC, both for women who went into labor and for those who had elective cesarean sections. This result can be explained, at least in part, by: the wide knowledge of the law in relation to the presence of a companion during delivery—Federal Law No. 11.108/2005, which allows women to recognize such violations of their rights [41]; and the feeling of fear, loneliness and vulnerability, which triggers in women the need for support from someone they trust at the time of delivery. In agreement with the results of this study, the research Birth in Brazil also identified the presence of a companion as a significant inhibitor of mistreatment [15].

Another factor present in the definition of the MLWC was not understanding the information given by health professionals, a situation that reduces the woman's autonomy to

participate in decisions about her care and makes her feel uncomfortable to clarify doubts. Protagonism in childbirth is associated with satisfaction regarding assistance [18, 42]; however, research shows that most women do not feel at ease during care. A study conducted in public maternity hospitals in a northeastern Brazilian state, for example, showed that 29.8% of women were dissatisfied with receiving guidance and the possibility of asking questions, and 49.8% were dissatisfied with the possibility of making complaints about the care provided [20]. Besides not having access to effective communication with health professionals, to be able to express their wishes and beliefs, many women suffer mistreatment and even abandonment, as a form of coercion, for being seen as complaining and demanding beings [1, 7].

The gap between the perception of violence by women and what is considered as the definition of this phenomenon, according to technical-scientific publications, is one of the most relevant aspects of this study. The data depicted on S1 Table allows us to identify several practices that fit the definition contained in national and international documents guiding childbirth care, but which, in the analysis carried out in this study, did not contribute with psychometric information in preparation of the instruments to measure MLWC. This may have occurred for two reasons: non-identification of the intervention or situation as violent by the women; and insufficient sample size to show significance in this model. The high number of women who answered "I don't know" or "I don't remember" to questions such as if labor was induced with oxytocin or if they'd rather have another position in labor likely reflects the lack of information about their rights, options and evidence base on intrapartum care. These data also points to communication gaps between birthing women and the health professionals assisting them.

Another point that deserves discussion is the feeling of unsafety reported by women, as well as their perception of not feeling welcomed: approximately one fourth of the participants in this study felt this way, and both aspects were items that reached high discrimination in the IRT analyses. These findings are extremely relevant, because they contrast in some way with the conception that the hospital is a safe environment for obstetric assistance. Our data show that not all women agree with this perspective. Previous studies have suggested that fear and unsafety during childbirth are linked with feelings of uncertainty, loss of control and interventions in their bodies [43, 44]. On the other hand, women who feel welcomed and perceive a positive environment, have a stronger sensation of safety [45]. Understanding more deeply the reasons why many women do not feel safe or welcomed in the hospital when they are giving birth is a matter that deserves further studies, given its importance in the assessment and measurement of mistreatment.

This study analyzes, in an unprecedented way, the mistreatment of women during childbirth using IRT to propose a way to measure this latent variable, through a set of items. The method, besides being statistically robust, evaluates every item of the measurement instrument according to its severity and discrimination capacity, defining for each one a different importance in estimating the MLWC. Moreover, the methodological rigor in conducting the study, with random selection, continuous quality control, and the timely face-to-face interviews also ensure higher quality to the study. Possible disadvantages of face-to-face interviews include embarrassment to address sensitive matters, such as criticizing the care received, more complex logistics, high costs, increased probability of non-participation (exclusions, losses and refusals). As an example, in our study, we had to exclude women living in areas with high occurrence of violent incidents, to preserve the safety of the research team. This represents a potential selection bias, because the women who live in these areas might be the most vulnerable ones, and their absence in the study sample makes the studied population somewhat different from the original one. In the same sense, women who were lost because of contact failure may be another source of selection bias. If we imagine that a more vulnerable profile implies in

higher risk of suffering mistreatment during childbirth [46, 47] then the inclusion of these women in the sample would probably enhance the discrimination capacity of the statistical model. However, the findings regarding the association between social vulnerability and mistreatment during childbirth are conflicting in the literature [42, 48].

Among the limitations of our study, one should consider the number of losses and the fact that the sample size is restricted to allow for more powerful conclusions about the set of items defined to measure the MLWC. Literature review points that to adjust an IRT model with high precision of parameter estimations, a sample of at least 500 is needed [49]. A smaller sample size represents a limitation in terms of precision, implying on less accurate estimates (with larger standard errors). However, as the objective of this study was to present a proposal for a measurement instrument and not a definitive instrument, we believe that the information obtained is relevant for the development of a final instrument, produced with larger samples and going through the process of its validation (of the instrument itself and the cutoff point) for the classification of women as to whether or not they have suffered mistreatment.

## Conclusion

The conclusions of this study are preliminary, considering that the model developed to measure the MLWC was based on a small sample. As a main finding, it was observed that the mistreatment level identified by IRT was approximately twice the general prevalence of the perception of disrespect, mistreatment or humiliation measured by the direct question to women, demonstrating the divergence between the perception of postpartum women and what is considered mistreatment by health agencies and academic literature. The IRT also made it possible to identify the items with greater severity and discrimination capacity, both for the general sample of women, and for the group that went into labor, pointing out situations that can be avoided in childbirth care, such as the absence of a companion, the lack of understanding of information by the woman, the performance of the fundal pressure maneuver, feelings of not being welcomed and of unsafety.

Harmonizing the speeches, knowledge and perceptions is essential to qualify obstetric care. The development of an instrument to measure mistreatment during childbirth from the perception of women is essential in this context, to standardize the prevalence measures in different national and international scenarios, thus allowing comparability and generalization, as well as to identify practices acknowledged as violent by women who go through the childbirth process and enable a dialogue between the different actors. This study can be considered a first step in this direction, making its own contribution to the proposal of strategies to eliminate obstetric violence.

## Supporting information

**S1 Fig. Flowchart of sample constitution.** Porto Alegre, 2016.
(TIF)

**S1 Dataset.**
(XLSX)

**S1 Table. Development of the IRT model from a set of situations and interventions usually considered in the definition of mistreatment of women in childbirth care.**
(DOCX)

**S2 Table. Description of the items of the measurement instruments for the Mistreatment Level of Women during Childbirth—MLWC1 (all women) and Mistreatment Level of**

**Women during Childbirth—MLWC2 (women who went into labor).**
(DOCX)

## Acknowledgments

The authors would like to thank Universidade Federal do Rio Grande do Sul and the group of researchers who took part in the project, in all its phases. The contribution of each one was vital to the study.

## Author Contributions

**Conceptualization:** Janini Cristina Paiz, Elsa Regina Justo Giugliani, Camila Giugliani.

**Data curation:** Janini Cristina Paiz, Stela Maris de Jezus Castro, Sarah Maria dos Santos Ahne, Camila Giugliani.

**Formal analysis:** Janini Cristina Paiz, Stela Maris de Jezus Castro, Elsa Regina Justo Giugliani, Camila Giugliani.

**Investigation:** Janini Cristina Paiz, Elsa Regina Justo Giugliani, Camila Giugliani.

**Methodology:** Janini Cristina Paiz, Stela Maris de Jezus Castro, Elsa Regina Justo Giugliani, Sarah Maria dos Santos Ahne, Camila Bonalume Dall' Aqua, Alice Steglich Souto, Camila Giugliani.

**Project administration:** Elsa Regina Justo Giugliani.

**Resources:** Elsa Regina Justo Giugliani.

**Software:** Janini Cristina Paiz, Stela Maris de Jezus Castro, Sarah Maria dos Santos Ahne, Camila Giugliani.

**Supervision:** Camila Bonalume Dall' Aqua, Alice Steglich Souto, Camila Giugliani.

**Writing – original draft:** Janini Cristina Paiz, Stela Maris de Jezus Castro, Sarah Maria dos Santos Ahne, Camila Bonalume Dall' Aqua, Camila Giugliani.

**Writing – review & editing:** Janini Cristina Paiz, Stela Maris de Jezus Castro, Elsa Regina Justo Giugliani, Sarah Maria dos Santos Ahne, Camila Bonalume Dall' Aqua, Alice Steglich Souto, Camila Giugliani.

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
