## [Decision Letter · Decision Letter 0]

14 Dec 2021

PONE-D-21-31080Development of an instrument to measure obstetric violence through Item Response TheoryPLOS ONE

Dear Dr. Paiz,

Thank you for submitting your manuscript to PLOS ONE. After careful consideration, we feel that it has merit but does not fully meet PLOS ONE’s publication criteria as it currently stands. Therefore, we invite you to submit a revised version of the manuscript that addresses the points raised during the review process.

We look forward to receiving your revised manuscript.

Kind regards,

Carla Betina Andreucci, M.D., P.h.D.

Academic Editor

PLOS ONE

Journal Requirements:

a) Did participants provide their written or verbal informed consent to participate in this study?

3. We note that you have stated that you will provide repository information for your data at acceptance. Should your manuscript be accepted for publication, we will hold it until you provide the relevant accession numbers or DOIs necessary to access your data. If you wish to make changes to your Data Availability statement, please describe these changes in your cover letter and we will update your Data Availability statement to reflect the information you provide

Reviewers' comments:

Reviewer's Responses to Questions

**Comments to the Author**

1. Is the manuscript technically sound, and do the data support the conclusions?

Reviewer #1: Yes

Reviewer #2: Yes

Reviewer #3: Partly

2. Has the statistical analysis been performed appropriately and rigorously? 

Reviewer #1: Yes

Reviewer #2: I Don't Know

Reviewer #3: I Don't Know

3. Have the authors made all data underlying the findings in their manuscript fully available?

Reviewer #1: No

Reviewer #2: Yes

Reviewer #3: Yes

4. Is the manuscript presented in an intelligible fashion and written in standard English?

Reviewer #1: No

Reviewer #2: No

Reviewer #3: No

5. Review Comments to the Author

Reviewer #1: Dear authors,

The manuscript "Development of an instrument to measure obstetric violence through Item Response Theory" brings significant and innovative knowledge to the study of obstetric violence as a public health problem. The discipline context and the research questions are well described, and the development of a tool to measure obstetric violence occurrence in maternity services is a relevant initiative.

I would like to suggest that you check the language (maybe you could hire a professional proofreader). I would also like to suggest some additional work in the manuscript, as follows.

1) Please describe the settings in which studied women gave birth. I suppose the two facilities are different in many aspects, for example, number of live births per year, health professionals usually responsible for (or involved in) childbirth care, training or teaching activities, c-section rates, etc. Additionally, those who do not know the Brazilian health system and the Brazilian typical obstetric care may find it difficult to understand the public/private contrast.

2) I understand the motivation to exclude women "who had formal contraindication for breastfeeding" and those who lived in places considered risky by the research team. However, I wonder if those women are not the most vulnerable ones, and therefore, the most likely to be subjected to obstetric violence. To my view, this is a potential source of bias that should be addressed in the study.

3) "Each day, two women from the public maternity hospital and one from the private hospital were included in the study until the intended sample was reached" (lines 124-126): please, what was the intended sample? How was it calculated? I am afraid I missed this information, and it is essential to discuss the study's limitations.

4) All information was collected during face to face interviews? How was socioeconomic level assessed? Please explain the categories' grouping.

5) Maybe you could attach a supplementary figure, or table, in order to clarify the number of women drawn in each hospital, the ones who were eligible, included, interviewed, etc. This would be an opportunity to clarify non-participation at each stage, for instance, the refusals and women who were excluded because they lived in a territory assessed as risky by the research team (it is not clear when or how this evaluation and exclusion was made).

6) Please, clarify why there are missing data (table 1).

7) I would strongly suggest that you add a table with more detailed information about women's outcomes, such as c-section rates, according to key characteristics, such as by type of hospital (public x private), women's skin color, etc. From previous studies we can suppose these results will have an impact in women's perception of mistreatment. Additionally, it is not clear if women who went into labor but had a c-section were included in the "women who went into labor" group or not.

8) In the same direction, I would suggest that you add some categories in table 1, for instance, women who did not use pain relief methods. Did they feel they were subjected to obstetric violence? I am eager to know that! Moreover, it was a bit difficult to me to understand why some categories showed were recommended practices ("encouraged to walk", "yes"), while others were the practices that should be abolished ("enema", "yes").

9) In the discussion section, I would suggest to include some thoughts on the data collection strategy. The face to face interview made outside the health facility may have advantages and disadvantages.

10) I would suggest you explore more deeply the feelings of unsafety reported by women. The hospitalization of childbirth care is often justified as the historical "evolution" of the obstetric assistance, understanding it as a pathway to safer care for women and babies. However, it seems that women do not agree with this. Additionally, we suppose that the health facility exists to provide care, but what does it mean when those who should be cared for say they don't feel welcomed? These reflections could reinforce the need for further studies on obstetric violence and the possibilities to measure it.

11) Maybe you could use the STROBE checklist before submitting the revised manuscript.

Reviewer #2: This cross-sectional study has a great potential to contribute to maternal health measures, as authors propose an instrument with 11 important components for quality of care.

According to Strobe Checklist, Introduction and Abstract are adequate. Methods sections would be improved if authors include a brief statement of any efforts carried on to address potential sources of bias; and also if they describe how data was collected, registered and organized. Results ans discussion are well described.

I suggest some background information update as the manuscript refers to "Rede Cegonha" which is now not available anymore, as it is not a public policy but a governamental program/project. Also, the findings that led to Itens exclusion as described on Supplementary Table 1 are very interesting and maybe authors could consider including one paragraph about those itens on the manuscript results and discussion, as they bring along interventions and situations that are highly prevalent in the country.

I'm affraid I couldn't check for the quality of the instrument development through the IRT Model, but the authors use literature fairly. I also suggest some minor revision on English overall, in order to improve understanding for readers.

Reviewer #3: This is a very important study for research on maternity care and reproductive health. However, some adjustments are necessary for this article to be published:

1. Please consider having the article proofread by an English expert so that we can ensure a better understanding and standardization of the text and the terms.

2. "Humanized" and "obstetric violence" are not the terms used by international researchers. I understand that these are terms used mainly in Brazil and Latin America, but considering that this article will be read by international readers, and that there are already well-known terms for both, it is important to use them or explain to the reader why you chose to use this definition instead of the well-established one.

3.Although I am not an expert in quantitative studies, it seems to me that the statistical aspects are well described. On the other hand, I missed more details about the questionnaire in the Data Collection section.

4. Lines 322 to 336 are presenting Results instead of Discussion. Consider moving up.

6. PLOS authors have the option to publish the peer review history of their article (what does this mean?). If published, this will include your full peer review and any attached files.

Reviewer #1: No

Reviewer #2: **Yes: **Ana Carolina Arruda Franzon

Reviewer #3: No

---

## [Author Response · Author response to Decision Letter 0]

6 Feb 2022

Dear Dr. Andreucci and reviewer,

We were happy to receive your feedback about our manuscript entitled Development of an instrument to measure obstetric violence through Item Response Theory [PONE-D-21-31080] and we are thankful for the careful reviews that have been carried out. We have thoroughly and carefully reviewed the manuscript and handled all of the points, one by one. We are sure to be now resubmitting to your appreciation an improved version of our article.

Please find below the point-by-point descriptions and responses to the reviews provided. Each point is addressed with a response in this letter and a description of the amendments that have been made to the manuscript and where in the text they can be viewed. In the manuscript, all amendments are marked in yellow for prompt identification.

Ethics statement has been amended, as well as data availability.

As suggested by all three reviewers, language has been checked by a professional (see attached Declaration) and many changes were made to the text.

Comments by Reviewer #1:

1. Please describe the settings in which studied women gave birth. I suppose the two facilities are different in many aspects, for example, number of live births per year, health professionals usually responsible for (or involved in) childbirth care, training or teaching activities, c-section rates, etc. Additionally, those who do not know the Brazilian health system and the Brazilian typical obstetric care may find it difficult to understand the public/private contrast.

Response: We have included the descriptions of the hospitals in the 1st paragraph under Study Design and Population (lines 124-132).

2. I understand the motivation to exclude women "who had formal contraindication for breastfeeding" and those who lived in places considered risky by the research team. However, I wonder if those women are not the most vulnerable ones, and therefore, the most likely to be subjected to obstetric violence. To my view, this is a potential source of bias that should be addressed in the study.

Response: Concerning the women who had formal contraindication for breastfeeding, we have made it clearer in the text that this was an elegibility criterion (not an exclusion one), because in the original research brestfeeding prevalence was a main outcome. This clarification was made in the 2nd paragraph under Study Design and Population (Lines 134-138). Concerning the women who lived in risky areas, that were actually excluded, we have included a new passage and added references to discuss this as a potential source of bias (before last paragraph of the Discussion – lines: 462-472).

3. "Each day, two women from the public maternity hospital and one from the private hospital were included in the study until the intended sample was reached" (lines 124-126): please, what was the intended sample? How was it calculated? I am afraid I missed this information, and it is essential to discuss the study's limitations.

Response: We have included a new paragraph (1st paragraph under Sample and Data Collection: lines 143-146) with information about sample size and calculation, with the

citation of the former research that originated the present study. We have also included in the last paragraph of the Discussion more information about the sample size needed to adjust an IRT model, with a new reference. This way, the study’s limitations concerning the sample size are clarified.

4. All information was collected during face to face interviews? How was socioeconomic level assessed? Please explain the categories' grouping.

Response: All information was collected during face to face interviews. This has been made clearer in the text, in the third paragraph under Sample and Data Collection (lines 160-165). In this same paragraph, we have explained how socioeconomic level was assessed and it’s categories grouped.

5. Maybe you could attach a supplementary figure, or table, in order to clarify the number of women drawn in each hospital, the ones who were eligible, included, interviewed, etc. This would be an opportunity to clarify non-participation at each stage, for instance, the refusals and women who were excluded because they lived in a territory assessed as risky by the research team (it is not clear when or how this evaluation and exclusion was made).

Response: A flowchart was included as supplementary Figure 1, cited in the manuscript in the 1st paragraph under Results (line 243).

6. Please, clarify why there are missing data (table 1).

Response: The reasons for missing data were clarified with footnotes in Table 1.

7. I would strongly suggest that you add a table with more detailed information about women's outcomes, such as c-section rates, according to key characteristics, such as by type of hospital (public x private), women's skin color, etc. From previous studies we can suppose these results will have an impact in women's perception of mistreatment. Additionally, it is not clear if women who went into labor but had a c-section were included in the "women who went into labor" group or not.

Response: The women who went into labor but had a c-section were included in the "women who went into labor" group. This information was made clearer in the second paragraph under Results. Also, Table 2 was added, presenting c-section rates and proportion of women who went into labor according to key characteristics. The findings of this new table are described in the Results session and some of them are discussed in a whole new paragraph in the Discussion (6th paragraph under that session), with new reference citations.

8. In the same direction, I would suggest that you add some categories in table 1, for instance, women who did not use pain relief methods. Did they feel they were subjected to obstetric violence? I am eager to know that! Moreover, it was a bit difficult to me to understand why some categories showed were recommended practices ("encouraged to walk", "yes"), while others were the practices that should be abolished ("enema", "yes").

Response: We have included in Table 1 the complementary category of each variable related to the care of women who went into labor. We believe this clarifies the issue, making it easier to understand those interventions and situations in light of being recommended or not.

9. In the discussion section, I would suggest to include some thoughts on the data collection strategy. The face to face interview made outside the health facility may have advantages and disadvantages.

Response: We have included some thoughts on potential disadvantages of face to face interviews in the before last paragraph of the Discussion (lines: 459-462).

10. I would suggest you explore more deeply the feelings of unsafety reported by women. The hospitalization of childbirth care is often justified as the historical "evolution" of the obstetric assistance, understanding it as a pathway to safer care for women and babies. However, it seems that women do not agree with this. Additionally, we suppose that the health facility exists to provide care, but what does it mean when those who should be cared for say they don't feel welcomed? These reflections could reinforce the need for further studies on obstetric violence and the possibilities to measure it.

Response: A new paragraph was added in the Discussion, with new reference citations, exploring this issue. This is the 10th paragraph under Discussion (lines: 441-452).

11. Maybe you could use the STROBE checklist before submitting the revised manuscript.

Response: The STROBE checklist was used and added as a supplementary file.

Comments by Reviewer #2:

1. According to Strobe Checklist, Introduction and Abstract are adequate. Methods sections would be improved if authors include a brief statement of any efforts carried on to address potential sources of bias; and also if they describe how data was collected, registered and organized. Results and discussion are well described.

Response: We have included a new paragraph describing how potential sources of bias were addressed, as well as included more details about data collection, registering and organization. This is the 5th paragraph under Sample and Data Collection (lines 171-178).

2. I suggest some background information update as the manuscript refers to "Rede Cegonha" which is now not available anymore, as it is not a public policy but a governamental program/project.

Response: We have reviewed updated information about “Rede Cegonha” and other programs related to prenatal and childbirth care in Brazil. We have modified the 4th paragraph of the Introduction (Lines 75-94) and inserted new sentences with updated references. In this review process we were able to certify that “Rede Cegonha” is still available and adherent institutions send their indicators monthly for monitoring. The ordinance supporting the program today is: Consolidation Ordinance n.3, 28 September 2017 (available at:

https://bvsms.saude.gov.br/bvs/saudelegis/gm/2017/prc0003_03_10_2017.html#ANEXOIITITIICAPII). Even though the program still exists, it has clearly weakened in terms of institutional strength, so we have included this information to clarify the context.

3. Also, the findings that led to Itens exclusion as described on Supplementary Table 1 are very interesting and maybe authors could consider including one paragraph about those itens on the manuscript results and discussion, as they bring along interventions and situations that are highly prevalent in the country.

Response: We have drawn more attention to episiotomy and amniotomy, which are very frequent practices in our country. We have included new passages in the 3rd paragraph under Results and in the 5th paragraph under Discussion. Also, the paragraph beginning with “The gap between the perception of violence” mentions some of the findings of Supplementary Table 1.

Comments by Reviewer #3:

1. "Humanized" and "obstetric violence" are not the terms used by international researchers. I understand that these are terms used mainly in Brazil and Latin America, but considering that this article will be read by international readers, and that there are already well-known terms for both, it is important to use them or explain to the reader why you chose to use this definition instead of the well-established one.

Response: we have reviewed these terms throughout the whole text. The term “obstetric violence” was replaced with “mistreatment of women during childbirth”. The term “level of obstetric violence” was replaced with “mistreatment level of women during childbirth”. Therefore the acronym NVO changed into MLWC.

2. Although I am not an expert in quantitative studies, it seems to me that the statistical aspects are well described. On the other hand, I missed more details about the questionnaire in the Data Collection section.

Response: We have provided more details about the questionnaire in the 3rd paragraph under Sample and Data Collection (lines 158-160).

3. Lines 322 to 336 are presenting Results instead of Discussion. Consider moving up.

Response: These lines were moved up to the Results session (lines: 256-259) and some remaining passages in the Discussion were adjusted accordingly.

We believe we have fully addressed all the points raised by the reviewers and we thank you again for the opportunity of qualifying our manuscript.

Yours sincerely, the author.

---

## [Decision Letter · Decision Letter 1]

26 Apr 2022

PONE-D-21-31080R1Development of an instrument to measure mistreatment of women during childbirth through Item Response TheoryPLOS ONE

Dear Dr. Paiz,

Thank you for submitting your manuscript to PLOS ONE. After careful consideration, we feel that it has merit but does not fully meet PLOS ONE’s publication criteria as it currently stands. Therefore, we invite you to submit a revised version of the manuscript that addresses the points raised during the review process.

The reviewers pointed out nomenclature issues (violence versus mistreatment), and aspects related to the recently revoking of the Stork Network in Brazil (Rede Cegonha). I suggest you include this debate in your manuscript.

We look forward to receiving your revised manuscript.

Kind regards,

Carla Betina Andreucci, M.D., P.h.D.

Academic Editor

PLOS ONE

Journal Requirements:

Reviewers' comments:

Reviewer's Responses to Questions

**Comments to the Author**

1. If the authors have adequately addressed your comments raised in a previous round of review and you feel that this manuscript is now acceptable for publication, you may indicate that here to bypass the “Comments to the Author” section, enter your conflict of interest statement in the “Confidential to Editor” section, and submit your "Accept" recommendation.

Reviewer #1: All comments have been addressed

Reviewer #2: All comments have been addressed

Reviewer #3: All comments have been addressed

2. Is the manuscript technically sound, and do the data support the conclusions?

Reviewer #1: Yes

Reviewer #2: Yes

Reviewer #3: Yes

3. Has the statistical analysis been performed appropriately and rigorously? 

Reviewer #1: I Don't Know

Reviewer #2: Yes

Reviewer #3: I Don't Know

4. Have the authors made all data underlying the findings in their manuscript fully available?

Reviewer #1: Yes

Reviewer #2: Yes

Reviewer #3: Yes

5. Is the manuscript presented in an intelligible fashion and written in standard English?

Reviewer #1: Yes

Reviewer #2: Yes

Reviewer #3: Yes

6. Review Comments to the Author

Reviewer #1: Dear authors,

Thank you for submitting your revised manuscript. As far as I am concerned, you improved the overall aspects of your paper and met the reviewers' requests.

I would like to bring up a few more questions just to deepen the discussion around obstetric violence scholarship. Indeed, I believe "obstetric violence" has a concept of its own, and it has gained ground in academic studies from various disciplines in the last decade, as can be verified from the growing number of articles published using this term. In my view, the use of "obstetric violence" poses a political stance, as it implies the recognition of structural problems that negatively affect women's experiences, jeopardizing their human rights.

I understand you changed to "mistreatment" to meet one of the reviewer's request, however, I was wondering which term or concept you used in the Brazilian Portuguese questionnaire, as I feel there may be differences in understanding between them. Unfortunately, I couldn't access the file "database_02_22.xlsx” to verify this. In any case, I would suggest that you reflect about the use of terms in your empirical material and in the manuscript before making a final decision between "obstetric violence" and "mistreatment".

I would like to point out two more specific issues for your consideration.

1) You mention that "All information considered in this study was collected during face-to-face interviews and the responses were referred by the participants. Women who were not found for the interview, after at least three attempts of contact by telephone and one in person, were considered a loss". However, in the results, you state that "The women not interviewed due to losses and refusals differed in terms of education and skin color, showing less education (p<0.01) and a higher prevalence of white skin color compared to those interviewed (p=0.032)". How did you get information about education and skin color from women who refused to participate?

2) Table 1, you added the information that "*Missing data correspond to responses 'I don’t know' or 'I don’t remember'" – for some variables, would it be relevant to consider these answers? For example, for "Induction with oxytocin", when women don't know whether or not they underwent this procedure, is it possible to state that they didn't receive appropriate information? Can we say the same about "would like another position in labor"? If 54 women answered "I don't know" or "I don't remember", it means that more than 25% of the birthing women didn't receive adequate information about mobility during labor – however, this variable was excluded from the model because of the high number of missing data. I don't mean that you have to re-process your data, but then again, I wonder what does it mean when we face such a high level of women who declare that they do not know if they would like another position in labor. It seems to me that they lack information about their rights, options and about evidence, as well.

Thank you once again for the opportunity to review your manuscript, I hope my comments were useful in some way.

All the best.

Reviewer #2: Dear authors and Editor, I'm pretty much satisfied with author's response to last review. I believe the manuscript was strengthened with the content that was added.

I believe an editorial and authorial decision will be need in order to comprehend or not the very recently published "PORTARIA GM/MS Nº 715, DE 4 DE ABRIL DE 2022" by Brazilian Ministry of Health, which alters the Rede Cegonha program into Rede de Atenção Materna e Infantil (Rami).

Reviewer #3: (No Response)

7. PLOS authors have the option to publish the peer review history of their article (what does this mean?). If published, this will include your full peer review and any attached files.

Reviewer #1: No

Reviewer #2: No

Reviewer #3: No

---

## [Author Response · Author response to Decision Letter 1]

19 Jun 2022

Dear Dr. Andreucci,

We were happy to receive your second feedback about our manuscript entitled Development of an instrument to measure obstetric violence through Item Response Theory [PONE-D-21-31080] and we are thankful for the careful reviews that have been carried out. We have once more handled all of the points, and we are thus resubmitting to your appreciation an improved version of our article.

Please find below the point-by-point descriptions and responses to the reviews provided. Each point is addressed with a response in this letter and a description of the amendments that have been made to the manuscript and where in the text they can be viewed. In the manuscript, all amendments are marked in yellow for prompt identification.

Comments by Reviewer #1:

1. I would like to bring up a few more questions just to deepen the discussion around obstetric violence scholarship. Indeed, I believe "obstetric violence" has a concept of its own, and it has gained ground in academic studies from various disciplines in the last decade, as can be verified from the growing number of articles published using this term. In my view, the use of "obstetric violence" poses a political stance, as it implies the recognition of structural problems that negatively affect women's experiences, jeopardizing their human rights.

I understand you changed to "mistreatment" to meet one of the reviewer's request, however, I was wondering which term or concept you used in the Brazilian Portuguese questionnaire, as I feel there may be differences in understanding between them. Unfortunately, I couldn't access the file "database_02_22.xlsx” to verify this. In any case, I would suggest that you reflect about the use of terms in your empirical material and in the manuscript before making a final decision between "obstetric violence" and "mistreatment".

Response: We are totally aligned with the view of this Reviewer. Although we have decided to keep the term “mistreatment” in the majority of the text, we have included the term “obstetric violence” in a few points, such as in the abstract, in order to point to the wider context reflected by this term, its poltical stance and the recognition of the structural issues involved. We have also included the term “obstetric violence” where the original study cited used this same term on its data collection. Our decision to keep the term “mistreatment” was for the following reasons: In the Brazilian Portuguese questionnaire we used the following question: “Have you ever (during labor and childbirth care) felt disrespected, humiliated or mistreated by health professionals?”, we have not actually used the term “obstetric violence”; the term mistreatment is more frequent in the international literature, so the use of this term can increase access and visibility of the article. Considering the relevance of the term “obstetric violence”, we decided to add a paragraph in the Introduction (lines 71-77).

2. You mention that "All information considered in this study was collected during face-to-face interviews and the responses were referred by the participants. Women who were not found for the interview, after at least three attempts of contact by telephone and one in person, were considered a loss". However, in the results, you state that "The women not interviewed due to losses and refusals differed in terms of education and skin color,

showing less education (p<0.01) and a higher prevalence of white skin color compared to those interviewed (p=0.032)". How did you get information about education and skin color from women who refused to participate?

Response: Name, age, education and skin color were drawn from the hospital chart, so these were the only data available for losses and refusals. We modified the text for clarification (Lines 169-171).

3. Table 1, you added the information that "*Missing data correspond to responses 'I don’t know' or 'I don’t remember'" – for some variables, would it be relevant to consider these answers? For example, for "Induction with oxytocin", when women don't know whether or not they underwent this procedure, is it possible to state that they didn't receive appropriate information? Can we say the same about "would like another position in labor"? If 54 women answered "I don't know" or "I don't remember", it means that more than 25% of the birthing women didn't receive adequate information about mobility during labor – however, this variable was excluded from the model because of the high number of missing data. I don't mean that you have to re-process your data, but then again, I wonder what does it mean when we face such a high level of women who declare that they do not know if they would like another position in labor. It seems to me that they lack information about their rights, options and about evidence, as well.

Response: We agree with the reviewer's suggestion and have therefore included a paragraph in the manuscript to briefly discuss these findings (lines 444-448).

Comments by Reviewer #2:

1. I believe an editorial and authorial decision will be need in order to comprehend or not the very recently published "PORTARIA GM/MS Nº 715, DE 4 DE ABRIL DE 2022" by Brazilian Ministry of Health, which alters the Rede Cegonha program into Rede de Atenção Materna e Infantil (Rami).

Response: We have included the recent “portaria 715/2022”, which alters the Rede Cegonha program, briefly presenting this still very recent context change (lines: 92-101).

We believe we have fully addressed all the points raised by the reviewers in this new round and we thank you again for the opportunity of qualifying our manuscript.

Yours sincerely, the authors.

---

## [Editor Report · Decision Letter 2]

28 Jun 2022

Development of an instrument to measure mistreatment of women during childbirth through Item Response Theory

PONE-D-21-31080R2

Dear Dr. Paiz,

We’re pleased to inform you that your manuscript has been judged scientifically suitable for publication and will be formally accepted for publication once it meets all outstanding technical requirements.

Kind regards,

Carla Betina Andreucci, M.D., P.h.D.

Academic Editor

PLOS ONE

---

## [Editor Report · Acceptance letter]

4 Jul 2022

PONE-D-21-31080R2 

Development of an instrument to measure mistreatment of women during childbirth through Item Response Theory 

Dear Dr. Paiz:

I'm pleased to inform you that your manuscript has been deemed suitable for publication in PLOS ONE. Congratulations! Your manuscript is now with our production department. 

Kind regards, 

on behalf of

Mrs. Carla Betina Andreucci 

Academic Editor

PLOS ONE